# Multipotent Mesenchymal Stem Cell-Based Therapies for Spinal Cord Injury: Current Progress and Future Prospects

**DOI:** 10.3390/biology12050653

**Published:** 2023-04-26

**Authors:** Chih-Wei Zeng

**Affiliations:** 1Department of Molecular Biology, University of Texas Southwestern Medical Center, Dallas, TX 75390, USA; chih-wei.zeng@utsouthwestern.edu; 2Hamon Center for Regenerative Science and Medicine, University of Texas Southwestern Medical Center, Dallas, TX 75390, USA

**Keywords:** multipotent mesenchymal stem cells, spinal cord injury, neuroprotection, neuronal regeneration, angiogenesis, regenerative medicine

## Abstract

**Simple Summary:**

Mesenchymal stem cells (MSCs) are a promising option for developing new treatments for spinal cord injury (SCI). They can help repair damaged tissue, making them an exciting area of research in regenerative medicine. It is important to study the safety, effectiveness, and best ways to use MSC-based therapies while addressing challenges in bringing these treatments to the clinic. Challenges include finding the best source of MSCs, determining when and how to administer them, and creating standardized methods for handling MSCs. Future research should focus on understanding the long-term effects of MSC treatments, optimizing their delivery, and conducting more clinical trials. Combining MSCs with other treatments might also improve outcomes for patients with spinal cord injuries. By increasing our knowledge of MSCs and their potential, we can offer hope for better recovery and quality of life for those affected by SCI.

**Abstract:**

Spinal cord injury (SCI) represents a significant medical challenge, often resulting in permanent disability and severely impacting the quality of life for affected individuals. Traditional treatment options remain limited, underscoring the need for novel therapeutic approaches. In recent years, multipotent mesenchymal stem cells (MSCs) have emerged as a promising candidate for SCI treatment due to their multifaceted regenerative capabilities. This comprehensive review synthesizes the current understanding of the molecular mechanisms underlying MSC-mediated tissue repair in SCI. Key mechanisms discussed include neuroprotection through the secretion of growth factors and cytokines, promotion of neuronal regeneration via MSC differentiation into neural cell types, angiogenesis through the release of pro-angiogenic factors, immunomodulation by modulating immune cell activity, axonal regeneration driven by neurotrophic factors, and glial scar reduction via modulation of extracellular matrix components. Additionally, the review examines the various clinical applications of MSCs in SCI treatment, such as direct cell transplantation into the injured spinal cord, tissue engineering using biomaterial scaffolds that support MSC survival and integration, and innovative cell-based therapies like MSC-derived exosomes, which possess regenerative and neuroprotective properties. As the field progresses, it is crucial to address the challenges associated with MSC-based therapies, including determining optimal sources, intervention timing, and delivery methods, as well as developing standardized protocols for MSC isolation, expansion, and characterization. Overcoming these challenges will facilitate the translation of preclinical findings into clinical practice, providing new hope and improved treatment options for individuals living with the devastating consequences of SCI.

## 1. Introduction

Spinal cord injuries (SCIs) are life-altering events with far-reaching consequences for both the individual and society. Each year, thousands of people suffer from SCI, often resulting in permanent disability, loss of independence, and a decreased quality of life [1]. Traditional treatment options have focused on rehabilitation and symptom management, but there remains a pressing need for innovative approaches to promote functional recovery and improve outcomes for SCI patients [2]. Advances in the field of regenerative medicine have led researchers to explore the potential of stem cells as a promising therapeutic strategy for spinal cord injury [3]. Among the various types of stem cells, multipotent mesenchymal stem cells (MSCs) have gained particular attention due to their ability to differentiate into multiple cell types, including those of the nervous system [4]. These cells can be sourced from various tissues, such as bone marrow, adipose tissue, and umbilical cord blood, making them a versatile option for potential therapies [5]. The application of MSCs in SCI treatment is based on their capacity to modulate the local environment, encourage tissue repair, and replace damaged cells [6]. As our understanding of the molecular mechanisms underlying MSC-mediated repair continues to grow, so too does the potential for their use in the development of novel therapies for SCI patients [7].

Recent studies offer a comprehensive analysis of current advancements in MSC research, emphasizing their potential to revolutionize SCI treatment [8]. The exploration of the molecular mechanisms behind MSC-mediated tissue repair highlights the versatility of these cells, which play a crucial role in various therapeutic processes, such as neuroprotection, neuronal regeneration, angiogenesis, immunomodulation, axonal regeneration, and glial scar reduction [9]. This multifaceted approach demonstrates the ability of MSCs to address the numerous challenges associated with SCIs, making them a promising candidate for future treatment strategies [10,11]. Researchers are also working to translate preclinical findings into real-world clinical applications for SCI patients [12]. Moreover, cell transplantation, tissue engineering, and cell-based therapies, such as MSC-derived exosomes, can harness the regenerative potential of MSCs to facilitate recovery and improve patients’ quality of life [13,14]. These therapeutic strategies hold immense promise for revolutionizing current treatment options and offering new hope for SCI patients [15]. However, there are challenges and limitations that must be overcome to bring MSC-based therapies into widespread clinical use [16]. These include optimizing the source, timing, and delivery methods of MSCs, as well as establishing standardized protocols for their isolation, expansion, and characterization [17,18]. Additionally, researchers must continue to assess the long-term safety and efficacy of MSC-based therapies to ensure their successful implementation in clinical settings [19].

In conclusion, the growing body of research on MSCs and their potential role in SCI treatment offers a valuable overview of the current state of the field. By examining the molecular mechanisms by which MSCs promote repair and exploring their potential clinical applications, researchers showcase the promise of MSC-based therapies for improving the lives of those affected by SCI [20]. As research progresses, it is essential to continue refining our understanding of MSCs, optimizing therapeutic strategies, and addressing the challenges that remain in translating these promising findings into routine clinical practice. By emphasizing the various molecular pathways and therapeutic strategies to which MSCs can contribute, the scientific community highlights the exciting future of MSC-based therapies for spinal cord injury patients while acknowledging the challenges that must be addressed to ensure their successful clinical translation.

## 2. Therapeutic Mechanisms of Mesenchymal Stem Cells in Spinal Cord Injury

Stem cells can be categorized based on their differentiation potential and developmental stages. Differentiation potential classifications include totipotent, pluripotent, multipotent, and unipotent cells, while developmental stage categories consist of embryonic, fetal, infant or umbilical cord blood, and adult stem cells (Figure 1). Among these, MSCs have garnered considerable interest as a promising candidate for SCI treatment due to their ability to differentiate into various cell types. Recent research has highlighted the multiple molecular mechanisms through which MSCs can promote recovery following SCI. In this discussion, we delve into the diverse functions MSCs perform in SCI repair and examine the specific molecular components involved.

### 2.1. Neuroprotection

MSCs secrete a variety of growth factors and cytokines that exhibit neuroprotective effects, playing a pivotal role in supporting the recovery of damaged neurons after SCI [21]. Among these factors are vascular endothelial growth factor (VEGF), nerve growth factor (NGF), insulin-like growth factor-1 (IGF-1), and brain-derived neurotrophic factor (BDNF), all of which contribute to a supportive microenvironment for neuronal survival and regeneration [22].

VEGF, for instance, not only promotes angiogenesis but also exerts direct neuroprotective effects by inhibiting apoptosis, reducing oxidative stress, and promoting neurogenesis [23]. NGF, on the other hand, supports the survival and growth of neurons, particularly those in the peripheral nervous system (PNS), by binding to its receptors, TrkA and p75NTR, and activating intracellular signaling pathways that promote neuronal survival [24]. IGF-1 contributes to neuroprotection by promoting neuronal survival, synaptic plasticity, and neurogenesis [25]. It has been shown to reduce inflammation, inhibit neuronal apoptosis, and stimulate the proliferation and differentiation of neural progenitor cells [26]. BDNF, another critical neurotrophic factor secreted by MSCs, enhances neuronal survival and function by activating the TrkB receptor and downstream signaling pathways, such as the PI3K/Akt and MAPK/ERK pathways [27].

Furthermore, MSCs can also modulate the expression of pro-inflammatory cytokines, such as tumor necrosis factor-alpha (TNF-α) and interleukin-1 beta (IL-1β), in microglia and astrocytes, and promote the release of anti-inflammatory cytokines like interleukin-10 (IL-10) and transforming growth factor-beta (TGF-β) from MSCs themselves as well as from other cell types, such as microglia and macrophages [28]. This modulation of the inflammatory milieu creates a more favorable environment for neuronal survival and recovery [29,30]. Overall, the neuroprotective effects of MSCs are multifaceted and involve a complex interplay of various growth factors and cytokines that work together to support the survival and regeneration of damaged neurons after SCI.

### 2.2. Promoting Neuronal Regeneration

MSCs possess the remarkable ability to differentiate into various neural cell types, including neurons and glial cells, such as astrocytes and oligodendrocytes [31,32]. Recent studies have shown that the differentiation of MSCs towards neurons or glial cells is orchestrated by a complex interplay of signaling pathways, transcription factors, and epigenetic modifications. For instance, key signaling pathways implicated in the neural differentiation of MSCs include the Notch, Wnt, and BMP signaling pathways [33,34]. Moreover, transcription factors such as Sox2, Pax6, and Neurogenin-2 play essential roles in guiding MSC differentiation toward neuronal and glial lineages [35,36,37]. The ratio of neurons to glial cells originating from MSCs is determined by the specific combination of signaling molecules and transcription factors present in the local microenvironment, which can be modulated by various extrinsic cues and experimental conditions. For example, the presence of growth factors like epidermal growth factor (EGF) and fibroblast growth factor (FGF) can promote neuronal differentiation [38,39], while the addition of ciliary neurotrophic factor (CNTF) can drive glial differentiation. The delicate balance of these factors ultimately influences the cell fate of MSCs and their potential to contribute to neural regeneration following CNS injury.

The pro-inflammatory niche within the SCI lesion can indeed affect the differentiation of MSCs. For instance, inflammatory cytokines, such as TNF-α and IL-1β, have been reported to influence MSC differentiation, potentially biasing MSCs towards a particular cell phenotype, such as astrocytes. High levels of these pro-inflammatory cytokines have been shown to inhibit neuronal differentiation while promoting the differentiation of MSCs into astrocytes [40,41]. On the other hand, anti-inflammatory cytokines, such as IL-4 and IL-10, have been reported to promote neuronal differentiation and inhibit glial differentiation of MSCs [42]. Importantly, MSCs also possess immunomodulatory properties, which may help mitigate inflammation and create a more favorable environment for tissue repair and regeneration [43,44]. MSCs can secrete various anti-inflammatory factors, such as TGF-β and IL-10, as well as modulate the function of immune cells, such as T cells and macrophages, to reduce inflammation and create a more permissive environment for tissue repair [45,46]. These MSC-mediated immunomodulatory effects can potentially counteract the negative influence of the pro-inflammatory niche on MSC differentiation, thus supporting their therapeutic potential in SCI.

Most studies have demonstrated that the preferential differentiation of MSCs towards a particular cell phenotype in the context of the pro-inflammatory niche within the SCI lesion could be a potential hurdle in using MSCs for post-SCI tissue regeneration. For example, excessive differentiation of MSCs into astrocytes, potentially exacerbated by the overexpression of transcription factors like STAT3, may contribute to glial scar formation, impeding axonal regeneration and functional recovery [47,48]. On the other hand, insufficient differentiation into oligodendrocytes, possibly hindered by the presence of inhibitory factors such as chondroitin sulfate proteoglycans (CSPGs), might limit remyelination and restoration of neuronal connectivity [49,50]. To address these challenges, it is crucial to gain a better understanding of the factors that influence MSC differentiation in the SCI environment and develop strategies, such as genetic modifications or controlled release of growth factors, to direct MSC differentiation towards the desired cell types. For instance, overexpressing transcription factors like Olig2 in MSCs can enhance their differentiation into oligodendrocytes and promote remyelination [51,52], while engineering MSCs to secrete specific growth factors, such as brain-derived neurotrophic factor (BDNF) or glial cell-derived neurotrophic factor (GDNF), can support neuronal survival and regeneration [53,54]. These examples highlight the importance of understanding and controlling MSC differentiation to maximize their therapeutic potential in post-SCI tissue regeneration.

The mechanical properties of extracellular matrix (ECM), such as stiffness, topography, and 3D architecture, can significantly affect MSC differentiation [55]. MSCs sense and respond to their mechanical environment through mechanotransduction, which involves converting mechanical signals into biochemical and cellular responses [56]. For instance, studies have shown that MSCs cultured on softer substrates with a stiffness similar to that of the brain tend to differentiate into neural lineages, whereas those cultured on stiffer substrates resembling bone tissue preferentially differentiate into osteogenic lineages [57]. Additionally, the topography and 3D architecture of the ECM can guide MSC alignment, migration, and differentiation by providing physical cues that influence cell shape and cytoskeletal organization [58,59]. In the context of SCI, optimizing the mechanical properties of the ECM could potentially enhance the therapeutic efficacy of MSCs by promoting their differentiation into the desired neural cell types and improving their integration with the host tissue.

This unique characteristic enables MSCs to replace damaged neural tissue, promote the regeneration of neuronal circuits, and ultimately contribute to functional recovery after SCI [27]. The process of neuronal regeneration is facilitated by the secretion of several trophic factors, such as BDNF, GDNF, NGF, and CNTF, which stimulate the growth, differentiation, and survival of neural cells [60]. Additionally, MSCs can promote the activation and proliferation of endogenous neural stem cells (NSCs) and progenitor cells within the injured spinal cord, further enhancing the regenerative process [61]. While it is clear that MSCs can differentiate into the component parts of neural circuits, the evidence for their ability to reestablish the correct neural circuits that existed prior to injury is still emerging. A few studies have reported that MSCs, either directly or indirectly, can contribute to the formation of functional neural circuits after SCI. For instance, Zeng et al. (2015) demonstrated that MSCs transplanted into the injured spinal cord were able to differentiate into neurons and form synapses with host neurons, contributing to the restoration of motor function [62]. Another study by Nakajima et al. (2012) showed that MSC transplantation promoted the growth of host corticospinal tract axons and the formation of new synapses [63].

Another crucial aspect of MSC-mediated neuronal regeneration involves the modulation of ECM components, such as CSPGs and matrix metalloproteinases (MMPs) [64,65]. By regulating the balance between ECM deposition and degradation, MSCs can create a more permissive environment for axonal growth and neural regeneration [66]. MSCs can also exert paracrine effects, which involve the release of extracellular vesicles (EVs) containing various bioactive molecules, such as proteins, lipids, and nucleic acids [67]. These EVs can transfer their cargo to recipient cells in the injured spinal cord, influencing their gene expression, proliferation, and differentiation, ultimately contributing to neuronal regeneration [68].

In summary, MSCs promote neuronal regeneration through various mechanisms, including their capacity to differentiate into neural cell types, the secretion of trophic factors, stimulation of endogenous NSCs, modulation of ECM components, and paracrine effects. These concerted actions of MSCs contribute to the restoration of neuronal circuits and functional recovery following SCI.

### 2.3. Angiogenesis

MSCs play a critical role in promoting the formation of new blood vessels by secreting pro-angiogenic factors, such as VEGF, angiopoietin-1, and basic fibroblast growth factor (bFGF) [69,70]. The process of angiogenesis is crucial for the recovery of injured spinal cord tissue, as it improves blood supply, accelerates tissue repair, and supports the survival of neural cells [71,72]. In addition to the secretion of pro-angiogenic factors, MSCs can also modulate the expression of various cell adhesion molecules and integrins, such as intercellular adhesion molecule-1 (ICAM-1) and vascular cell adhesion molecule-1 (VCAM-1), which facilitate the recruitment and migration of endothelial cells to the injury site [73,74]. This process contributes to the formation of new blood vessels and enhances the overall regenerative capacity of the injured spinal cord [75].

MSC-derived EVs also contribute to the angiogenic process by transferring bioactive molecules, such as microRNAs (miRNAs) and proteins, to the recipient endothelial cells [76]. These transferred molecules can regulate gene expression, promote endothelial cell proliferation, migration, and tube formation, ultimately stimulating angiogenesis in the injured spinal cord [77]. Additionally, MSCs can establish communication with other cell types, such as pericytes and astrocytes, which play essential roles in the stabilization and maturation of newly formed blood vessels [78]. By engaging in crosstalk with these cells, MSCs can ensure the proper development and functionality of the newly generated vascular network within the injured spinal cord [79].

In summary, MSCs contribute to angiogenesis through various mechanisms, including the secretion of pro-angiogenic factors, modulation of cell adhesion molecules, release of EVs, and interaction with other cell types involved in vascular development. These collective actions of MSCs help improve blood supply to the injured spinal cord, facilitate tissue repair, and support neural cell survival, ultimately contributing to functional recovery after SCI.

### 2.4. Immunomodulation

MSCs possess remarkable immunomodulatory properties that contribute to their therapeutic potential in SCI treatment [80]. Their ability to modulate the activity of various immune cells, such as macrophages, T-cells, B-cells, and natural killer (NK) cells, helps control inflammation, prevent autoimmune responses, and create a more favorable environment for tissue repair [81,82]. MSCs can regulate the polarization of macrophages, promoting a switch from the pro-inflammatory M1 phenotype to the anti-inflammatory M2 phenotype [83]. This shift in macrophage polarization is essential for controlling inflammation and fostering an environment that supports tissue repair and regeneration [84]. Additionally, MSCs can suppress the activation and proliferation of T-cells, modulate their cytokine secretion profile, and induce the generation of regulatory T-cells (Tregs), which play a crucial role in maintaining immune tolerance and preventing autoimmune responses [85,86]. MSCs can also inhibit B-cell activation, proliferation, and antibody production, further dampening the potential for harmful immune reactions [87].

MSCs can directly interact with NK cells, downregulating their cytotoxic activity and pro-inflammatory cytokine production [44,88]. Moreover, MSCs can secrete various soluble factors, such as TGF-β, prostaglandin E2 (PGE2), and indoleamine 2,3-dioxygenase (IDO), which contribute to their immunomodulatory effects [89]. Another important aspect of MSC-mediated immunomodulation is the release of EVs, which contain bioactive molecules, such as proteins, lipids, and nucleic acids [90,91]. These EVs can mediate intercellular communication and modulate the function of recipient immune cells, thus contributing to the overall immunomodulatory effects of MSCs [92].

In summary, MSCs exert their immunomodulatory effects through various mechanisms, including the modulation of immune cell polarization, secretion of soluble factors, and release of EVs. These actions help control inflammation, prevent autoimmune responses, and create a more favorable environment for tissue repair and regeneration following SCI.

### 2.5. Axonal Regeneration

MSCs play a crucial role in promoting axonal regeneration after SCI through various mechanisms, including the secretion of diverse neurotrophic factors, such as NGF, CNTF, and FGF [93,94,95]. These factors not only stimulate the growth of new axons but also support the survival and differentiation of neurons, ultimately leading to improved connectivity and functionality in the injured spinal cord [96]. In addition to secreting neurotrophic factors, MSCs can influence the local cellular environment by releasing cytokines and chemokines that recruit endogenous stem cells to the site of injury [97,98]. This recruitment of stem cells can further support axonal regeneration and promote tissue repair by providing additional cellular resources for the formation of new neuronal connections [99].

MSCs can also promote axonal regeneration by directly interacting with neurons and fostering the extension of growth cones [100]. This interaction can be mediated by various cell adhesion molecules and extracellular signaling molecules, such as N-cadherin and ephrin family members, which help guide axonal growth and encourage the formation of new synaptic connections [101]. Moreover, MSCs can enhance the intrinsic growth capacity of injured neurons, including both central nervous system (CNS) and PNS neurons, such as dorsal root ganglion (DRG) neurons, by upregulating the expression of regeneration-associated genes (RAGs), such as growth-associated protein-43 (GAP-43), arginase-1 (Arg-1), and activating transcription factor-3 (ATF3) [102,103,104,105]. These RAGs play a critical role in the regenerative process by supporting the growth and guidance of axons and promoting synaptic plasticity [106]. Furthermore, MSCs can form cellular bridges at the injury site, which help guide regenerating axons across the lesion and re-establish connections with target neurons [107]. This scaffold-like structure created by MSCs can enhance the overall regenerative capacity of the injured spinal cord, leading to improved functional recovery.

In summary, MSCs promote axonal regeneration through multiple mechanisms, including the secretion of neurotrophic factors, recruitment of endogenous stem cells, direct interaction with neurons, upregulation of regeneration-associated genes, and formation of cellular bridges. These combined actions contribute to enhanced axonal growth and improved functional recovery after SCI.

### 2.6. Glial Scar Reduction

Glial scar formation is a natural response to SCI, characterized by the activation of astrocytes and the deposition of various ECM components [108]. Although glial scars play a protective role in limiting the spread of inflammation and tissue damage, they also create a physical and biochemical barrier to axonal regeneration, ultimately impeding functional recovery [48].

MSCs can modulate glial scar formation by targeting multiple aspects of this process. Firstly, they can regulate the activation and proliferation of astrocytes by secreting anti-inflammatory cytokines, such as interleukin-4 (IL-4) and interleukin-13 (IL-13) [109]. These cytokines inhibit the pro-inflammatory signaling pathways that drive astrocyte activation and proliferation, thereby limiting glial scar formation [110,111]. Secondly, MSCs can directly influence the production and degradation of various ECM components within the glial scar. By secreting ECM-modulating enzymes, MSCs can regulate the balance of ECM synthesis and breakdown, ensuring an optimal remodeling process that supports axonal regeneration [112,113]. Additionally, MSCs have been reported to modulate the expression of genes involved in ECM synthesis, such as fibronectin and laminin [114,115], and genes involved in ECM degradation, such as MMPs and tissue inhibitors of metalloproteinases (TIMPs) [116,117]. These MSC-mediated mechanisms impact the composition of the glial scar and influence its permissiveness for axonal growth. Moreover, MSCs can promote the infiltration of macrophages and microglia into the glial scar, which can help facilitate the clearance of inhibitory debris and ECM components [118]. This process is supported by the secretion of chemoattractants, such as C-X-C motif chemokine ligand 12 (CXCL12), chemokine (C-C motif) ligand 2 (CCL2), and platelet-derived growth factor (PDGF), which recruit these immune cells to the site of injury [119,120]. Furthermore, MSCs can alter the phenotype of reactive astrocytes, driving them toward a more permissive state that supports neuronal regeneration [60]. This phenotypic switch can be mediated by the secretion of factors such as insulin-like growth factor-1 (IGF-1), FGF-2, and erythropoietin (EPO), which promote the expression of genes associated with tissue repair and axonal growth [121,122].

In summary, MSCs play a vital role in promoting neuronal regeneration through a variety of mechanisms (Table 1). Firstly, their ability to differentiate into neural cell types, such as neurons and glial cells, enables them to replace damaged tissue and contribute to repair. Secondly, they secrete trophic factors like BDNF, GDNF, NGF, and CNTF, promoting growth, differentiation, and survival of neural cells and fostering a supportive regeneration environment. Thirdly, MSCs enhance regeneration by stimulating endogenous NSCs and progenitor cells within the injured spinal cord, leading to the formation of new neural cells. Fourthly, they modulate ECM components, including CSPGs and MMPs, balancing ECM deposition and degradation to create a permissive environment for axonal growth and neural regeneration. Finally, MSCs exert paracrine effects through the release of EVs containing bioactive molecules such as proteins, lipids, and nucleic acids. These EVs transfer their cargo to recipient cells in the injured spinal cord, influencing gene expression, proliferation, and differentiation, ultimately contributing to neuronal regeneration. Collectively, these diverse mechanisms employed by MSCs collaborate to restore neuronal circuits and enhance functional recovery following SCI. Exploiting these actions can significantly improve the therapeutic potential of MSCs for treating spinal cord injuries and other CNS disorders.

## 3. Clinical Applications of Multipotent Stem Cells in SCI Treatment

The unique molecular mechanisms of MSCs render them a promising candidate for the development of innovative therapies for SCI. In this discussion, we explore the potential clinical applications of MSCs in the treatment of SCI. We also provide a detailed comparison of the three MSC-based clinical applications in SCI treatment, highlighting their advantages and challenges in a more comprehensive manner.

### 3.1. Cell Transplantation

Direct transplantation of MSCs into the injured spinal cord has emerged as a promising therapeutic strategy, with MSCs demonstrating the ability to differentiate into neural cells, modulate the local environment, and promote repair [123]. In addition to their regenerative potential, MSCs have also been shown to migrate to the site of injury, further highlighting their potential for targeted therapy [94,124]. Clinical trials have shown promising results, with a study reporting significant improvements in motor and sensory function following MSC transplantation [125]. In another clinical trial, patients treated with MSC transplantation showed improvements in motor, sensory, and autonomic functions, as well as a reduction in neuropathic pain [126]. Moreover, MSC transplantation has been associated with reduced inflammation, decreased glial scar formation, and enhanced axonal regeneration, all of which contribute to the overall functional recovery after SCI [127,128,129]. Despite these promising findings, there are several challenges that must be addressed to optimize MSC transplantation as a viable therapy for SCI. One such challenge is determining the optimal source of MSCs, as cells can be isolated from various tissues, such as bone marrow, adipose tissue, and umbilical cord blood, each with its unique advantages and limitations [130,131]. The timing of MSC transplantation is another important consideration, as the stage of injury and inflammatory response may significantly impact the therapeutic outcome [132,133].

Developing safe and effective delivery methods for MSC transplantation is also critical. Currently, researchers are exploring various routes of administration, such as intravenous, intrathecal, or intraspinal injections, to determine the most efficient and least invasive approach [134]. Furthermore, long-term safety and efficacy remain major concerns. Studies must continue to assess the potential risks associated with MSC transplantation, such as tumorigenicity [135,136], unwanted differentiation, or immune rejection [137,138]. By addressing these challenges and refining the MSC transplantation process, researchers will be better equipped to harness the regenerative potential of MSCs and develop effective therapeutic strategies for spinal cord injury patients.

### 3.2. Tissue Engineering

MSCs can be combined with biomaterial scaffolds to create engineered tissue constructs that closely resemble the native spinal cord structure [107]. These constructs can be implanted into the injured spinal cord, offering a supportive environment for tissue regeneration and functional recovery [107,139]. Scaffolds can be fabricated from natural materials, such as collagen, chitosan, or hyaluronic acid, or synthetic materials, such as poly(lactic-co-glycolic acid) (PLGA) and polyethylene glycol (PEG), each with specific properties that can be tailored to support MSC survival, differentiation, and integration into the host tissue [140,141]. Tissue engineering approaches can also incorporate the controlled release of growth factors or other bioactive molecules to enhance the regenerative potential of MSCs [142]. By incorporating these molecules into the scaffold, a sustained release can be achieved, promoting a more conducive environment for neural regeneration and functional recovery. Moreover, advances in microfabrication and 3D bioprinting technologies have enabled the development of more complex and biomimetic scaffold designs [143,144,145]. These technologies can create spatially defined microenvironments within the scaffold, allowing for the precise control of cell distribution and organization. This level of control has the potential to improve the formation of functional neural circuits and enhance the overall therapeutic outcome.

Despite the promising results seen in preclinical studies, further research is needed to optimize scaffold design [146], identify the most effective combination of MSCs and biomaterials [147], and evaluate the long-term safety and efficacy of these engineered constructs in clinical settings [148]. By addressing these challenges, tissue engineering approaches utilizing MSCs have the potential to significantly impact the treatment of spinal cord injuries, providing novel strategies to facilitate functional recovery and improve patients’ quality of life.

### 3.3. Cell-Based Therapies

MSC-derived exosomes have attracted significant attention as a cell-free therapeutic approach for SCI due to their regenerative and neuroprotective properties [149,150]. These small EVs are released by MSCs and contain a variety of bioactive molecules, including growth factors, signaling molecules, microRNAs, and proteins, which can contribute to tissue repair and functional recovery [151]. Exosome-based therapies offer several advantages over traditional cell transplantation methods. First, they reduce the risk of immune rejection and inflammatory responses, as exosomes are considered to be immunologically inert [152]. This makes them particularly suitable for allogeneic transplantation, where the donor and recipient are not genetically identical [153]. Second, exosome-based therapies are less likely to induce tumor formation or unwanted differentiation, as they do not contain live cells that could potentially proliferate uncontrollably or differentiate into undesired cell types [154,155]. This contributes to an improved safety profile for exosome-based treatments. Third, exosomes are more easily stored, transported, and administered compared to live cell products, making them a more feasible option for widespread clinical use [17,153]. Moreover, exosomes can be concentrated and purified, allowing for precise control over dosage and therapeutic potency [156].

Despite these advantages, there are several challenges that must be addressed before exosome-based therapies can be successfully implemented in clinical settings. Standardizing exosome isolation and characterization protocols is essential to ensure the reproducibility of results and the quality of exosome products [157,158]. Additionally, researchers must develop methods to efficiently deliver exosomes to the site of injury while maintaining their bioactivity and minimizing potential side effects. Finally, further studies are needed to evaluate the long-term safety and efficacy of exosome-based therapies in clinical trials. These trials will provide essential data on the therapeutic potential of MSC-derived exosomes in the treatment of SCI and help guide the development of future therapies based on this promising approach.

In conclusion, MSCs hold great promise for revolutionizing SCI treatment due to their unique regenerative properties. By exploring their potential in cell transplantation, tissue engineering, and cell-based therapies, we can work towards improving the lives of those affected by SCI. As research advances, it’s essential to deepen our understanding of MSCs, refine therapies, and tackle challenges in translating findings to clinical practice. This involves optimizing MSC sources, delivery methods, and intervention timing, as well as standardizing protocols for isolation, expansion, and characterization. Conducting well-designed clinical trials will help assess the safety, efficacy, and long-term outcomes of MSC-based therapies (Table 2). By addressing these challenges and building on current research, we can unlock the full potential of MSC-based therapies and create innovative treatments for SCI and other neurological disorders. This progress has the potential to greatly impact the lives of patients and their families, offering hope for a better future.

## 4. Challenges and Future Directions

While MSC-based therapies hold great potential for revolutionizing SCI treatment, numerous challenges must be addressed before they can be widely implemented in clinical settings. A key challenge is optimizing MSC sources, isolation, and culture methods to ensure the therapeutic efficacy and safety of the cells [159,160]. Identifying the optimal timing and delivery methods for MSC transplantation is crucial, as these factors greatly impact treatment outcomes. Long-term assessments of safety and efficacy are necessary to provide a comprehensive understanding of the potential risks and benefits of MSC-based therapies [161]. For patients with chronic SCI, MSC-based therapies may offer some hope for promoting functional recovery. Studies have reported positive effects of MSC transplantation in animal models of chronic SCI, including improvements in functional outcomes, reduced glial scar formation, and enhanced axonal regeneration [127,129]. However, the challenges associated with treating chronic SCI, such as the long-standing presence of inhibitory molecules, glial scar formation, and neuronal loss, necessitate further research to optimize MSC-based strategies for this patient population. It is important to explore innovative approaches, such as combinatorial therapies, targeted drug delivery, or tissue engineering techniques, to enhance the therapeutic potential of MSCs for chronic SCI patients. Deepening our understanding of the complex SCI microenvironment will help enhance the regenerative potential of transplanted MSCs and inform the development of targeted strategies [123,162,163]. Additionally, investigating the potential of combinatorial therapies, such as integrating MSC-based treatments with rehabilitation, pharmacological interventions, or electrical stimulation, may lead to synergistic effects and improved therapeutic outcomes [164].

Developing standardized protocols and guidelines for MSC-based therapies is essential to ensure consistency in treatment approaches and facilitate comparisons of outcomes across clinical trials. Addressing the regulatory and ethical challenges associated with MSC-based therapies is also crucial for responsible translation into clinical practice [165]. Moreover, future research should explore innovative therapeutic strategies, such as using MSC-derived exosomes, gene-modified MSCs, or tissue engineering approaches incorporating biomaterials and scaffolds, which may offer additional benefits and overcome some limitations of traditional MSC-based therapies. By tackling these challenges and investigating novel therapeutic avenues, the field of regenerative medicine can continue to advance and improve clinical outcomes for patients with SCI.

Recent studies have investigated the use of MSCs for treating CNS disorders, including not only SCI but also traumatic brain injury (TBI) [166,167,168]. However, it is important to understand the similarities and differences in the molecular and cellular pathogenesis of TBI and SCI as it relates to MSC-based therapy (Table 3). Although both TBI and SCI involve damage to the CNS and share common mechanisms, such as inflammation [169], glial scar formation [170], and neuronal apoptosis [171], they also exhibit distinct molecular and cellular pathogenesis. These differences impact the functional deficits experienced by patients, the specific cell types, and the neural circuits involved, as well as the optimal timing and delivery of MSC-based therapies. For instance, in TBI, MSCs may need to be delivered earlier to counteract the rapid spread of inflammation and tissue damage [172], while in SCI, their delivery might be more effective during the subacute phase to promote tissue regeneration and reduce scar formation [47,173]. Despite these differences, MSCs appear to have promising therapeutic potential for both conditions. However, when designing MSC-based therapies, it is crucial to consider the specific cellular and molecular context of each injury type to optimize their efficacy and safety. For example, the use of MSCs overexpressing specific growth factors, such as BDNF for TBI [174] or GDNF for SCI [175], may enhance their therapeutic effects by targeting the unique pathophysiological features of each injury.

## 5. Conclusions

Multipotent stem cells represent a promising direction for the development of innovative therapies for spinal cord injuries. The molecular mechanisms by which MSCs promote tissue repair and their potential clinical applications have generated significant interest and hope in the field of regenerative medicine. As research progresses, it is imperative to continue investigating the safety, efficacy, and optimal delivery methods of MSC-based therapies while addressing the challenges associated with translating preclinical findings into widespread clinical applications. Such challenges encompass determining the ideal source of MSCs, the most suitable timing for intervention, and the most effective route of administration. Additionally, it is crucial to establish standardized methods for isolating, expanding, and characterizing MSCs to ensure reproducibility and patient safety. Future studies should continue to explore the potential of MSCs and their derivatives in treating spinal cord injuries. This involves investigating the long-term effects of MSC transplantation, optimizing delivery methods, and conducting further clinical trials to establish the safety and efficacy of MSC-based therapies. Moreover, the exploration of combinatory therapies that involve MSCs with other treatment modalities, such as electrical stimulation or rehabilitative training, may further enhance the therapeutic outcomes for SCI patients. As our understanding of the molecular mechanisms underlying MSCs’ regenerative properties deepens, we move closer to realizing the full potential of this promising therapeutic approach for the millions of individuals living with the devastating consequences of spinal cord injuries. This progress offers renewed hope for improved recovery and quality of life for SCI patients and their families, ultimately transforming the landscape of SCI treatment and management.

## Figures and Tables

**Figure 1 biology-12-00653-f001:**
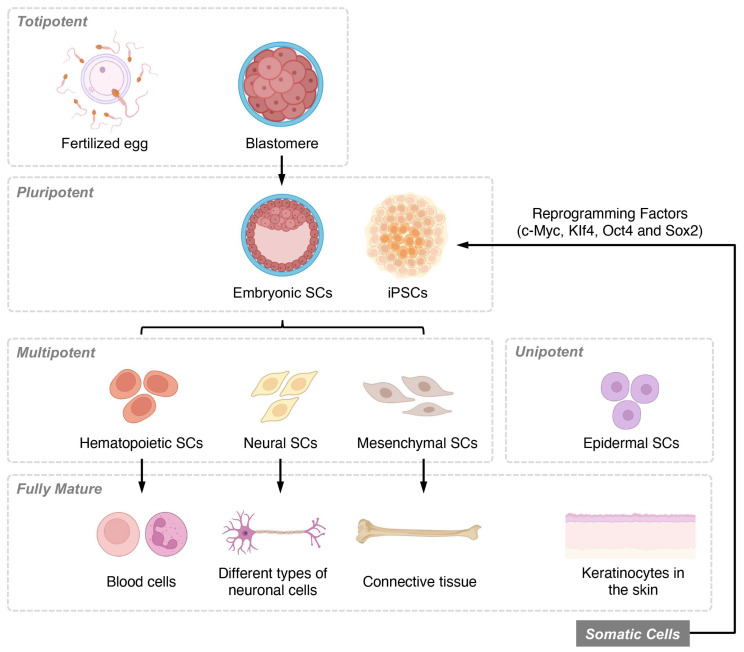
Overview of stem cell classifications based on differentiation potential and developmental stages. Stem cells are classified based on their differentiation potential, which reflects their ability to form various cell types. Totipotent stem cells, found in early-stage embryos, can differentiate into all three germ layers, as well as extra-embryonic tissues and placental cells. Pluripotent stem cells, present in the blastocyst stage of development, maintain the ability to self-renew and differentiate into the three germ layers and multiple lineages but cannot form extra-embryonic tissues or placental cells. Multipotent stem cells, also known as adult or somatic stem cells, are undifferentiated cells found in postnatal tissues. These specialized cells have limited self-renewal capabilities and are committed to specific lineages. Unipotent stem cells are the most restricted in their differentiation potential, as they can only give rise to a single cell type, although they still retain the capacity for self-renewal.

**Table 1 biology-12-00653-t001:** Therapeutic Mechanisms of Mesenchymal Stem Cells in Spinal Cord Injury.

Mechanism	Key Factors/Processes	Description	References
Neuroprotection	VEGF, NGF, IGF-1, BDNF, Anti-inflammatory cytokines	MSCs secrete growth factors and cytokines that support neuronal survival and regeneration, reduce inflammation, and create a supportive microenvironment for damaged neurons.	[21,22,23,24,25,27,29,30]
Promoting Neuronal Regeneration	Differentiation, Trophic factors, Activation of endogenous NSCs, ECM modulation, Paracrine effects	MSCs can differentiate into neural cell types, secrete trophic factors, stimulate endogenous NSCs, modulate ECM components, and exert paracrine effects, contributing to the regeneration of neuronal circuits and functional recovery.	[33,34,35,36,37,38,39]
Angiogenesis	Pro-angiogenic factors, Cell adhesion molecules, EVs, Interaction with other cell types	MSCs promote the formation of new blood vessels through the secretion of pro-angiogenic factors, modulation of cell adhesion molecules, release of EVs, and interaction with other cell types involved in vascular development.	[71,72,73,74,76,77]
Immunomodulation	Immune cell modulation, Soluble factors, EVs	MSCs modulate the activity of immune cells, secrete soluble factors, and release EVs to control inflammation, prevent autoimmune responses, and create a favorable environment for tissue repair and regeneration.	[44,81,82,83,85,86,87,88,89,90,91]
Axonal Regeneration	Neurotrophic factors, ECM modulation, Direct interaction with neurons, EVs, Regeneration-associated genes	MSCs secrete neurotrophic factors, modulate ECM, directly interact with neurons, release EVs, and upregulate RAGs to promote axonal regeneration, leading to enhanced axonal growth and improved functional recovery.	[93,94,95,97,98,100,101,102,103,104,105,106]
Glial Scar Reduction	Anti-inflammatory cytokines, ECM degradation, Infiltration of immune cells, Alteration of reactive astrocyte phenotype	MSCs modulate astrocyte activation, regulate ECM production and degradation, promote immune cell infiltration, and alter the phenotype of reactive astrocytes to reduce glial scar formation and create a permissive environment for regeneration.	[48,108,109,110,111,112,113,114,115,116,117,118,119,120,121,122]

**Table 2 biology-12-00653-t002:** Detailed comparison of MSC-based clinical applications in SCI treatment.

Clinical Application	Advantages	Challenges	References
Cell Transplantation	MSCs differentiate into neural cells, replacing damaged tissue	Identifying optimal MSC source (bone marrow, adipose tissue, umbilical cord blood)	[130,131]
	Modulation of local environment (immunomodulation, angiogenesis, axonal regeneration)	Determining the best timing of transplantation	[132,133]
	MSC migration to injury site for targeted therapy	Developing safe, effective delivery methods	[134]
	Improved motor, sensory, and autonomic functions in clinical trials	Long-term safety concerns (tumorigenicity, unwanted differentiation, immune rejection)	[135,136,137,138]
Tissue Engineering	Controlled release of growth factors or bioactive molecules	Evaluating long-term safety and efficacy in clinical settings	[142]
	Supportive environment for tissue regeneration	Optimizing scaffold design (natural vs. synthetic materials)	[146]
	Customizable scaffolds from natural or synthetic materials	Identifying the most effective combination of MSCs and biomaterials	[147]
	Advanced microfabrication and 3D bioprinting for biomimetic designs	-	[148]
Cell-based Therapies (MSC-derived exosomes)	Reduced risk of immune rejection and inflammatory responses	Standardizing exosome isolation and characterization protocols	[152]
	Lower likelihood of tumor formation or unwanted differentiation	Developing efficient delivery methods while maintaining bioactivity	[154,155]
	Easier storage, transport, and administration	Evaluating long-term safety and efficacy in clinical trials	[17,153]
	Concentration and purification for precise dosage control	-	[156]

**Table 3 biology-12-00653-t003:** Similarities and differences between TBI and SCI in the context of MSCs-based therapy.

	Traumatic Brain Injury (TBI)	Spinal Cord Injury (SCI)
Similarities	Direct mechanical forces (e.g., contusion, penetration)	Direct mechanical forces (e.g., compression, transection)
	Inflammation is a common mechanism	Inflammation is a common mechanism
	Glial scar formation	Glial scar formation
	Neuronal apoptosis	Neuronal apoptosis
	MSCs promote neuroprotection, angiogenesis, and immunomodulation	MSCs promote neuroprotection, angiogenesis, and immunomodulation
Differences	TBI primarily affects brain regions, leading to cognitive and emotional impairments	SCI primarily affects motor and sensory functions of the spinal cord
	Blood-brain barrier disruption is more prevalent in TBI	Blood-spinal cord barrier disruption is a concern in SCI
	MSCs may need to be targeted to specific brain regions in TBI	MSCs can be administered locally or systemically in SCI
	MSCs may need to promote neuronal circuit repair in TBI	MSCs may need to promote axonal regeneration in SCI
	MSCs may modulate TBI-induced neuroinflammation (ex: BDNF)	MSCs may modulate SCI-induced neuroinflammation (ex: GDNF)

## Data Availability

Data sharing is not applicable to this article.

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
