# Peer review of "Multipotent Mesenchymal Stem Cell-Based Therapies for Spinal Cord Injury: Current Progress and Future Prospects"

_biology, 2023, doi:10.3390/biology12050653_

Round 1

Reviewer 1 Report

Review by Chih-Wei Zeng “Multipotent Stem Cell-based Therapies for Spinal Cord Injury: Current Progress and Future Prospects” is timely and addresses important aspect of the modern translational neurosciences – novel cell-based therapies for spinal cord injury (SCI). The review covers broad range of the recent literature in unbiased manner.

However, in my opinion the manuscript will benefit from revision.

Please find below several comments than should be addressed:

First of all, I have a commentary regarding the terminology used in this article. The author uses abbreviation MSCs for multipotent stem cells, however, MSCs is an abbreviation commonly used for mesenchymal stem cells, and this may lead to consusion and misunderstanding. Often, while talking about multipotent cells in general author refers to publications describing data from experiments with mesenchymal stem cells.

Mesenchymal stem cells are multipotent stem cells, indeed, but not all multipotent stem cells are mesenchymal stem cells.

Line 144 - The author state that “MSCs possess the remarkable ability to differentiate into various neural cell types, including neurons and glial cells, such as astrocytes and oligodendrocytes“.

What are the molecular mechanisms regulating the differentiation of MSCs towards the neurons or glial cells and determining the ratio of neurons/glial cells originating from MSCs? In particular, will the pro-inflammatory niche within the SCI lesion shift the differentiation preferentially towards the particular cell phenotype? Can it be one of potential hurdles in terms of using MSCs for post-SCI tissue regeneration? How do mechanical characteristics of ECM affect MSCs differentiation?

Line 154 - “Another crucial aspect of MSC-mediated neuronal regeneration involves the modulation of extracellular matrix (ECM) components, such as chondroitin sulfate proteoglycans (CSPGs) and matrix metalloproteinases (MMPs) [34, 35]”.

The author quotes dissertation (reference 34), this dissertation is based on several original publications. Why does not the author refer to original publications instead?

Line 156 – The author states that “by regulating the balance between ECM deposition and degradation, MSCs can create a more permissive environment for axonal growth and neural regeneration [36]”, however reference number 36 is a review, not an original research article. The author should quote several original works to support the statement that MSCs regulate ECM deposition/degradation.

Line 265 – The author says “By secreting ECM-modulating enzymes, MSCs can regulate the balance of ECM synthesis and breakdown, ensuring an optimal remodeling process that supports axonal regeneration” [84], however, I could not find this information in the reference 84 [Chen, S. L., et al, 2015].

Line 267 – The author state that “MSCs can modulate the expression of genes involved in ECM synthesis and degradation, such as aggrecan (ACAN) [85], versican (VCAN) [86], and hyaluronan synthases (HAS) [87], impacting the composition of the glial scar and influencing its permissiveness for axonal growth”. Based on analysis of the quoted literature I tend to disagree with this statement. In my understanding, levels of ACAN were increased in chondrocytes under dynamic compression (according to [85]), but not via the MSCs-directed mechanism. Next, the reference [87] demonstrates that secretome of some cancer cell lines may induce HA-synthesis of bmMSCs, not vice versa.

Line 299 – The author state “Clinical trials have demonstrated encouraging results, with several studies reporting significant improvements in 300 motor and sensory function following MSC transplantation [95]”. However, reference [95] is only one clinical trial, not several clinical studies.

Importantly, the focus of this review is MSCs-based therapy for SCI, however many articles quoted by the author are focused on traumatic brain injury, not SCI. The author should highlight similarities and differences of TBI and SCI molecular and cellular pathogenesis, as applicable to MSCs-based therapy.

Reviewer 2 Report

Specific comments are attached in the pdf. Generally, this is a well written and clearly argued review. There are numerous incomplete reference citations that need amending.

This is a well written paper and only minor grammatical errors were detected

Round 2

Reviewer 1 Report

I am satisfied with the updated version of the manuscript, however I recommend removing new table 3 from the text.